# Shifting Mean Activation Towards Zero with Bipolar Activation Functions

## Abstract

We propose a simple extension to the ReLU-family of activation functions that allows them to shift the mean activation across a layer towards zero. Combined with proper weight initialization, this alleviates the need for normalization layers. We explore the training of deep vanilla recurrent neural networks (RNNs) with up to 144 layers, and show that bipolar activation functions help learning in this setting. On the Penn Treebank and Text8 language modeling tasks we obtain competitive results, improving on the best reported results for non-gated networks. In experiments with convolutional neural networks without batch normalization, we find that bipolar activations produce a faster drop in training error, and results in a lower test error on the CIFAR-10 classification task.

## 1 Introduction

Recurrent neural networks (RNN) are able to model complex dynamical systems, but are known to be hard to train (Pascanu et al., 2013b). One reason for this is the vanishing or exploding gradient problem (Bengio et al., 1994). Gated RNNs like the Long Short-Term Memory (LSTM) of Hochreiter & Schmidhuber (1997) alleviate this problem, and are widely used for this reason. However, with proper initialization, non-gated RNNs with Rectified Linear Units (ReLU) can also achieve competitive results (Le et al., 2015).

The choice of activation function has strong implications for the learning dynamics of neural networks. It has long been known that having zero-centered inputs to a layer leads to faster convergence times when training neural networks with gradient descent (Le Cun et al., 1991). When inputs to a layer have a mean that is shifted from zero, there will be a corresponding bias to the direction of the weight updates, which slows down learning (LeCun et al., 1998). Clevert et al. (2015) showed that a mean shift away from zero introduces a bias shift for units in the next layer, and that removing this shift by zero-centering activations brings the standard gradient closer to the natural gradient (Amari, 1998).

The Rectified Linear Unit (ReLU) (Nair & Hinton, 2010; Glorot et al., 2011) is defined as $f(x) = max(x, 0)$ and has seen great success in the training of deep networks. Because it has a derivative of 1 for positive values, it can preserve the magnitude of the error signal where sigmoidal activation functions would diminish it, thus to some extent alleviating the vanishing gradient problem. However, since it is non-negative it has a mean activation that is greater than zero.

Several extensions to the ReLU have been proposed that replace its zero-valued part with negative values, thus allowing the mean activation to be closer to zero. The Leaky ReLU (LReLU) (Maas et al., 2013) replaces negative inputs with values that are scaled by some factor in the interval $[0, 1]$. In the Parametric Leaky ReLU (PReLU) (He et al., 2015) this scaling factor is learned during training. Randomized Leaky ReLUs (RReLU) (Xu et al., 2015) randomly sample the scale for the negative inputs. Exponential Linear Units (ELU) (Clevert et al., 2015) replace the negative part with a smooth curve which saturates to some negative value.

Concurrent to our work, Klambauer et al. (2017) proposed the Scaled ELU (SELU) activation function, which also has self-normalizing properties, although it takes an orthogonal and complimentary approach to the one proposed here.

Chernodub & Nowicki (2016) proposed the Orthogonal Permutation Linear Unit (OPLU), where every unit belongs to a pair $\{x_i, x_j\}$, and the activation function simply sorts this pair:

$$\begin{pmatrix} z_i \\ z_j \end{pmatrix} = \begin{pmatrix} \max(x_i, x_j) \\ \min(x_i, x_j) \end{pmatrix}$$ (1)

This function has many desirable properties: It is norm and mean preserving, and has no diminishing effect on the gradient.

The Concatenated Rectified Linear Unit (CReLU) (Shang et al., 2016) concatenates the ReLU function applied to the positive and negated input $f'(x) = (f(x), f(-x))$. Balduzzi et al. (2017) combined the CReLU with a mirrored weight initialization $W_1 f(x) - W_2 f(-x)$, with $W_1 = W_2$ at initialization. The resulting function is initially linear, and thus mean preserving, before training starts.

Another approach to maintain a mean of zero across a layer is to explicitly normalize the activations. An early example was Le Cun et al. (1991), who suggested zero-centering activations by subtracting each units mean activation before passing to the next layer. In Glorot & Bengio (2010) it was shown that the problem of vanishing gradients in deep models can be mitigated by having unit variance in layer activations. Batch Normalization (Ioffe & Szegedy, 2015) normalizes both the mean and variance across a mini-batch. The success of Batch Normalization for deep feed forward networks created a research interest in how similar normalization of mean and variance can be extended to RNNs. Despite early negative results (Laurent et al., 2016), by keeping separate statistics per timestep and properly initializing parameters, Batch Normalization can be applied to the recurrent setting (Cooijmans et al., 2016). Other approaches to hidden state normalization include Layer Normalization (Ba et al., 2016), Weight Normalization (Salimans & Kingma, 2016) and Norm Stabilization (Krueger & Memisevic, 2015).

The Layer-Sequential Unit Variance (LSUV) algorithm (Mishkin & Matas, 2015) iteratively initializes each layer in a network such that each layer has unit variance output. If such a network can maintain approximately unit variance throughout training, it is an attractive option because it has no runtime overhead.

In this paper, we propose bipolar activation functions as a way to keep the layer activations approximately zero-centered. We explore the training of deep recurrent and feed forward networks, with LSUV-initialization and bipolar activations, without using explicit normalization layers like batch norm.

## 2 BIPOLAR ACTIVATION FUNCTIONS

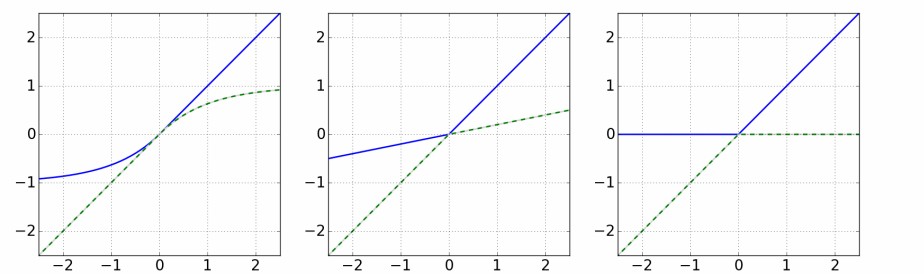

Figure 1: Bipolar versions of popular activation functions. From left: Bipolar ELU, Bipolar Leaky ReLU, Bipolar ReLU.

In a neural network, the ReLU only preserves its positive inputs, and thus shifts the mean activation in a positive direction. However, if we for every other neuron preserve the negative inputs, this effect

will be canceled for zero-centered i.i.d. input vectors. In general, for any ReLU-family activation function $f$, we can define its bipolar version as follows:

$$f_B(x_i) = \begin{cases} f(x_i), & \text{if } i \bmod 2 = 0 \\ -f(-x_i), & \text{if } i \bmod 2 \neq 0 \end{cases} \quad (2)$$

For convolutional layers, we flip the activation function in half of the feature maps.

**Theorem 1.** *For a layer of bipolar ReLU units, this trick will ensure that a zero-centered i.i.d. input vector x will give a zero-centered output vector. If the input vector has a mean different from zero, the output mean will be shifted towards zero.*

*Proof.* Let $x_1$ and $x_2$ be the input vectors to the ordinary units and the flipped units respectively. The output vectors from the two populations are $f_B(x_1) = max(0, x_1)$ and $f_B(x_2) = -f(-x_2) = min(0, x_2)$. Since $x$ is i.i.d. then $x$, $x_1$ and $x_2$ have the same distribution, and they have the same expectation value $\mathbb{E}[x] = \mathbb{E}[x_1] = \mathbb{E}[x_2]$. Then the expectation value of the output can be simplified

$\mathbb{E}[f_B(x)] = 0.5\mathbb{E}[f_B(x_1) + f_B(x_2)] = 0.5\mathbb{E}[max(0, x_1) + min(0, x_2)] = 0.5\mathbb{E}[max(0, x) + min(0, x)] = 0.5\mathbb{E}[x]$. $\square$

**Theorem 2.** *For a layer of bipolar ELU units, this trick will ensure that a i.i.d. input vector will give an output mean that is shifted towards a point in the interval $[-\alpha, \alpha]$, where $\alpha$ is the parameter defining the negative saturation value of the ELU.*

*Proof.* Let $x_1$ and $x_2$ be the input vectors to the ordinary units and the flipped units respectively. Let $x_1^+$ and $x_2^+$ be the vectors of positive values in the two populations, and let $x_1^-$ and $x_2^-$ be the negative values. The output vectors from the four populations are $f_B(x_1^+) = x_1^+$, $f_B(x_1^-) = \alpha(e^{x_1^-} - 1)$, $f_B(x_2^+) = \alpha(1 - e^{-x_2^+})$ and $f_B(x_2^-) = x_2^-$. Since $x$ is i.i.d. then $x$, $x_1$ and $x_2$ have the same distribution, and they have the same expectation value $\mathbb{E}[x] = \mathbb{E}[x_1] = \mathbb{E}[x_2]$. We also have that $\mathbb{E}[x_1^+] = \mathbb{E}[x_2^+]$ and $\mathbb{E}[x_1^-] = \mathbb{E}[x_2^-]$.

For some fraction of positive values $\beta \in [0, 1]$ we can write

$\mathbb{E}[f_B(x_1)] = \beta\mathbb{E}[f_B(x_1^+)] + (1 - \beta)\mathbb{E}[f_B(x_1^-)] = \mathbb{E}[\beta x_1^+ + (1 - \beta)\alpha(e^{x_1^-} - 1)]$ and

$\mathbb{E}[f_B(x_2)] = \beta\mathbb{E}[f_B(x_2^+)] + (1 - \beta)\mathbb{E}[f_B(x_2^-)] = \mathbb{E}[\beta\alpha(1 - e^{-x_2^+}) + (1 - \beta)x_2^-]$.

Then the expectation value of the output can be simplified

$\mathbb{E}[f_B(x)] = 0.5\mathbb{E}[f_B(x_1) + f_B(x_2)] =$

$0.5\mathbb{E}[\beta x_1^+ + (1 - \beta)\alpha(e^{x_1^-} - 1) + \beta\alpha(1 - e^{-x_2^+}) + (1 - \beta)x_2^-] =$

$0.5\mathbb{E}[x + (1 - \beta)\alpha(e^{x_1^-} - 1) + \beta\alpha(1 - e^{-x_2^+})] = 0.5\mathbb{E}[x + z]$.

We can see that $z$ is bounded in the interval $[-\alpha, \alpha]$. If $\mathbb{E}[x]$ is different from $\mathbb{E}[z]$, then the output mean $\mathbb{E}[f_B(x)]$ will be shifted towards the point $\mathbb{E}[z]$ inside the interval $[-\alpha, \alpha]$. $\square$

Theorem 1 says that for bipolar ReLU, an input vector $x$ that is not zero-centered, the mean will be pushed towards zero. Theorem 2 says that for bipolar ELU, an input vector $x$ will be pushed towards a value in the interval $[-\alpha, \alpha]$. These properties have a stabilizing effect on the activations.

Figure 2 shows the evolution of the dynamical system $x_{i+1} = f(Wx_i)$ for different activation functions $f$. As can be seen, the bipolar activation functions have more stable dynamics, less prone to exhibiting an exploding mean and variance.

## 3 DEEPLY STACKED RNNS

Motivated by the success of very deep models in other domains, we investigate the training of deeply stacked RNN models. As we shall see, certain problems arise that are unique to the recurrent setting.

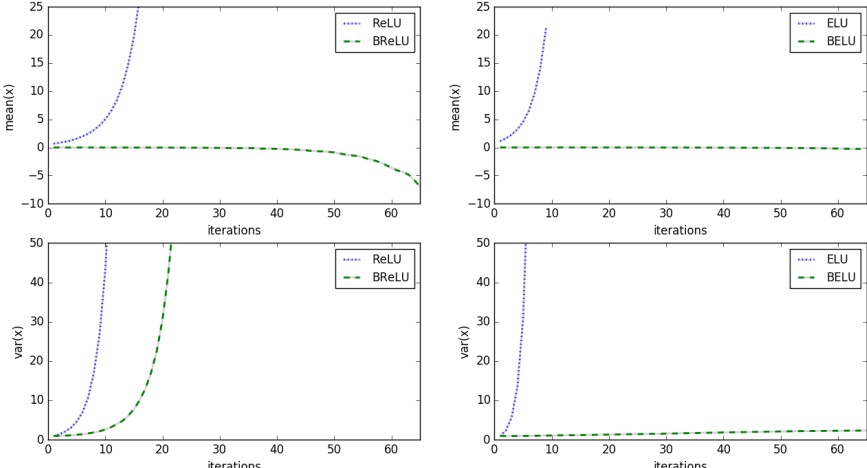

Figure 2: Iterations of $x_{i+1} = f(Wx_i)$ for different activation functions $f$ with $x_1 \sim \mathcal{N}(0,1)$ and $W \sim \mathcal{N}(0, \sigma^2)$, with $\sigma$ set by the LSUV procedure such that $f(Wx_1)$ has approximately unit variance. The graphs show the mean and variance of $x_i$, averaged over 50 separate runs, where a different $x_1$ and $W$ was sampled each run.

The network architecture we consider consists of stacks of vanilla RNN units (Elman, 1990), with the recurrent update equation for layer $i$:

$$h_i(t) = f(W_i h_i(t-1) + U_i x_i(t) + b_i) \tag{3}$$

For the first layer, we encode the input as a fixed embedding vector $x_1(t) \sim \mathcal{N}(0,1)$. Subsequent layers are fed the output of the layers below, $x_i(t) = h_{i-1}(t)$.

If each layer in a neural network scales its input by a factor $k$, the scale at layer L will be $k^L$. For $k \neq 1$ this leads to exponentially exploding or vanishing activation magnitudes in deep networks. Notice that this phenomenon holds for any path through the computational graph of an unrolled RNN, both for the within-timestep activation magnitudes across layers, and for the within-layer activation magnitudes across timesteps.

In order to avoid exploding or vanishing dynamics, we want to have unit variance on $h_i(t)$. To this end, we adapt the LSUV weight initialization procedure (Mishkin & Matas, 2015) to the recurrent setting by considering a single timestep of the RNN, and setting the input from the recurrent connections to be $\sim \mathcal{N}(0,1)$. The LSUV procedure is then simply to go through each layer sequentially, adjusting the magnitude of $W_i$ and $U_i$ to produce an output $h_i(t)$ with unit variance, while propagating new activations through the network after each weight adjustment.

Since $W_i$ and $U_i$ have the same magnitude at the start of the initialization procedure and are scaled in synchrony, they will have the same magnitude as each other when the initialization procedure is complete. This means that the input-to-hidden connections $U_i$ and the hidden-to-hidden connections $W_i$ contribute equal parts to the unit variance of $h_i(t)$, and thus that the gradient flows in equal magnitude across the horizontal and vertical connections. [1]

---

[1]After the LSUV initialization, it is possible to rescale $W_i$ and $U_i$ in order to explicitly trade off the extent to which the gradient should flow along each direction. By choosing a $\gamma \in [0,1]$, we can trade off what portion of the variance each matrix contributes, while maintaining the variance of the output of the layer:

$$W_i' = W_i \sqrt{2\gamma} \qquad \text{and} \qquad U_i' = U_i \sqrt{2 \cdot (1-\gamma)}$$

We note that this seems like a potent approach for influencing the time horizon at which an RNN should learn, but do not explore it further in this work. In our experiments $W_i$ and $U_i$ have equal magnitude at initialization (i.e. $\gamma = 0.5$).

While LSUV initialization allows training to work in deeper stacks of RNN layers, even with LSUV we get into trouble when the stacks get deep enough (see Appendix B). Visualizing the gradient flow reveals that while the gradient does flow from first layer to the last, it takes a diagonal path backwards through time. The effect is that the initial learning in layer $N$ is most strongly influenced by the input $N$ timesteps in the past.

We have included a brief discussion and visualization of this problem in the appendix, where it can be seen that this problem is remedied if we for every 4th layer $i = 4, 8, 12, 16...$ add a skip connection:

$$h_i(t) = f(W_i h_i(t-1) + U_i x_i(t) + b_i) + \alpha h_{i-4}(t) \tag{4}$$

The use of skip connections has been shown previously to aid learning in deeply stacked RNNs (Graves, 2013). Note that because of the LSUV init, we know that input from the skip connection $h_{i-4}(t)$ has approximately unit variance. Because of this, the initialization procedure could scale $W_i$ and $U_i$ down to zero, and still have unit variance $h_i(t)$. To avoid this effect, we scale down the skip connection slightly, setting $\alpha = 0.99$.

## 4 EXPERIMENTS

### 4.1 CHARACTER-LEVEL PENN TREEBANK

We train character level RNN language models on Penn Treebank (Marcus et al., 1993), a 6MB text corpus with a vocabulary of 54 characters. Because of the small size of the dataset, proper regularization is key to getting good performance.

The RNNs we consider follow the architecture described in the previous section: Stacks of simple RNN layers, with skip connections between groups of 4 layers, LSUV initialization, and inputs encoded as an $\mathcal{N}(0, 1)$ fixed embedding.

We seek to investigate the effect of *depth* and the effect of *bipolar activations* in such deeply stacked RNNs, and run a set of experiments to illuminate this.

The models were trained using the ADAM optimizer (Kingma & Ba, 2014), with a learning rate of 0.0002, a batch size of 128, on non-overlapping sequences of length 50. Since the training data does not cleanly divide by 50, for each epoch we choose a random crop of the data which does, as done in Cooijmans et al. (2016). We calculated the validation loss every 4th epoch, and divided the learning rate by 2 when the validation loss did not improve. No gradient clipping was used.

For regularization, we used a combination of various forms of dropout. We used standard dropout between layers, as done in Pham et al. (2013); Zaremba et al. (2014). On the recurrent connections, we followed *rnnDrop* (Moon et al., 2015) and used the same dropout mask for every timestep in a sequence. We adapted stochastic depth (Huang et al., 2016) to the recurrent setting, stochastically dropping entire blocks of 4 layers, replacing recurrent and non-recurrent connections with identity connections for a timestep (this was explored for single units, called *Zoneout*, and also on single layers in Krueger et al. (2016)). Unlike Huang et al. (2016), we did not do any rescaling of the droppable blocks at test time, since this would go against the goal of having unit variance on the output of each block.

We look at model depths in the set $\{4, 8, 12, 24, 36\}$ and on activation functions ReLU and ELU and their bipolar versions BReLU and BELU. Since good performance on this dataset is highly dependent on regularization, in order to get a fair comparison of various depths and activation functions, we need to find good regularization parameters for each combination. We have dropout on recurrent connections, non-recurrent connections and on blocks of four layers, and thus have 3 separate dropout probabilities to consider. In order to limit this large parameter search space, we first do an exploratory search on dropout probabilities in the set $\{0, 0.025, 0.05\}$, for the depths 4 and 36 and functions ReLU and BELU. For both recurrent connections dropout and block dropout, we get best results with 0.025 dropout probability (for every model). However, we find that the ideal between-layer dropout probability decreases with model depth. We therefore freeze all other parameters, and consider between-layer dropout probabilities in the set $\{0.025, 0.05, 0.1\}$, for ReLU networks in depths 4 and 36. We use the optimal probabilities for each depth found here for the other activation functions. For

the remaining depths we choose a probability between the best setting for 4 and 36 layers. For the best performing function at 36 layers (BELU), we also train deeper models of 48 and 144 layers.

To get results comparable with previously reported results, we constrained the number of parameters in each model to be about the same as for a network with 1x1000 LSTM units, approximately 4.75M parameters.

Table 1: Penn Treebank validation errors (BPC)

|  | dropout | #parameters | ReLU | BReLU | ELU | BELU |
|---|---|---|---|---|---|---|
| 4 x 760 | 0.1 | ~4.75M | 1.321 | 1.324 | DNC | 1.318 |
| 8 x 540 | 0.075 | ~4.75M | 1.331 | 1.320 | DNC | 1.317 |
| 16 x 385 | 0.075 | ~4.75M | 1.321 | 1.324 | DNC | 1.317 |
| 24 x 314 | 0.075 | ~4.75M | 1.349 | 1.334 | DNC | 1.317 |
| 36 x 256 | 0.05 | ~4.75M | 1.353 | 1.319 | DNC | 1.311 |
| 48 x 222 | 0.05 | ~4.75M | - | - | - | 1.320 |
| 144 x 128 | 0.025 | ~4.75M | - | - | - | 1.402 |

From Table 1 we can see that ReLU-RNN performed worse with increasing depth. With ELU-RNN, learning did not converge. The bipolar version of ELU avoids this problem, and its performance does not degrade with increasing depth up to 36 layers. Overall, the best validation BPC is achieved with the 36 layer BELU-RNN. Figure 6 shows the training error curves of the 36-layer RNN with each of the activation functions, and shows that the bipolar variants see a faster drop in training error.

We briefly explore substituting the SELU unit into the 36-layer RNN. There, as with the ELU, the training quickly diverges. This phenomenon occurs even with very low learning rates. However, if we substitute the SELU with its bipolar variant, training works again. A 36-layer BSELU-RNN converges to a validation error of 1.314 BPC, similar to BELU.

Some previously reported results on the test set is included in Table 2. The listed models have approximately the same number of parameters. We can see that the results for the 36 layer BELU-RNN is better than for the best reported results for non-gating architectures (DOT(S)-RNN), but also competitive with the best normalized and regularized LSTM architectures. Notably, the BELU-RNN outperforms the LSTM with recurrent batch normalization (Cooijmans et al., 2016), where both mean and variance are normalized.

Table 2: Penn Treebank test error

| Network | BPC |
|---|---|
| Tanh + Zoneout (Krueger et al., 2016) | 1.52 |
| ReLU 1x2048 (Neyshabur et al., 2016) | 1.47 |
| GRU + Zoneout (Krueger et al., 2016) | 1.41 |
| MI-RNN 1x2000 (Wu et al., 2016) | 1.39 |
| DOT(S)-RNN (Pascanu et al., 2013a) | 1.386 |
| LSTM 1x1000 (Krueger et al., 2016) | 1.356 |
| LSTM 1x1000 + Stoch. depth (Krueger et al., 2016) | 1.343 |
| LSTM 1x1000 + Recurrent BN (Cooijmans et al., 2016) | 1.32 |
| LSTM 1x1000 + Dropout (Ha et al., 2016) | 1.312 |
| LSTM 1x1024 + Rec. dropout (Semeniuta et al., 2016) | 1.301 |
| LSTM 1x1000 + Layer norm (Ha et al., 2016) | 1.267 |
| LSTM 1x1000 + Zoneout (Krueger et al., 2016) | 1.252 |
| Delta-RNN + Dropout (II et al., 2017) | 1.251 |
| HM-LSTM 3x512 + Layer norm (Chung et al., 2016) | 1.24 |
| HyperNetworks (Ha et al., 2016) | **1.233** |
| BELU 36x256 | 1.270 |

## 4.2  CHARACTER-LEVEL TEXT8

Text8 (Mahoney, 2011) is a simplified version of the Enwik8 corpus with a vocabulary of 27 characters. It contains the first 100M characters of Wikipedia from Mar. 3, 2006. We also here trained an RNN to predict the next character in the sequence. The dataset was split taking the first 90% for training, the next 5% for validation and the final 5% for testing, in line with common practice. The test results reported is the test error for the epoch with lowest validation error.

The network architecture here was identical to the 36 layer network used in the Penn Treebank experiments, except that we used a larger layer size of 474. We used an initial learning rate of 0.00005, which was halved when validation error did not improve from one epoch to the next. To match previously reported results, we constrained the number of parameters to be 16.2M, about the same as for a 1x2000 LSTM network. We chose 0.01 dropout probability for stochastic depth, recurrent and non-recurrent dropout, and did not do a hyperparameter search on this dataset.

On this dataset, training diverges with both ReLU and ELU due to exploding activation dynamics. These problems do not occur with their bipolar variants.

Table 3: Text8 validation error [BPC]

| Network | ReLU | BReLU | ELU | BELU |
|---|---|---|---|---|
| 36x474 RNN | DNC | 1.399 | DNC | 1.334 |

We compare the result on the test set with reported results obtained with approximately the same number of parameters. From Table 4 we can see that the result for the 36 layer BELU-RNN improves upon the best reported result for non-gated architectures (Skipping-RNN).

Table 4: Text8 test error

| Network | BPC |
|---|---|
| MI-Tanh 1x2048 (Wu et al., 2016) | 1.52 |
| LSTM 1x2048 (Wu et al., 2016) | 1.51 |
| Skipping-RNN (Pachitariu & Sahani, 2013) | 1.48 |
| MI-LSTM 1x2048 (Wu et al., 2016) | 1.44 |
| LSTM 1x2000 (Cooijmans et al., 2016) | 1.43 |
| LSTM 1x2000 (Krueger et al., 2016) | 1.408 |
| mLSTM 1x1900 (Krause et al., 2016) | 1.40 |
| LSTM 1x2000 + Recurrent BN (Cooijmans et al., 2016) | 1.36 |
| LSTM 1x2000 + Stochastic depth (Krueger et al., 2016) | 1.343 |
| LSTM 1x2000 + Zoneout (Krueger et al., 2016) | 1.336 |
| Recurrent Highway Network (Zilly et al., 2016) | **1.29** |
| HM-LSTM 3x1024 + Layer norm (Chung et al., 2016) | **1.29** |
| BELU 36x474 | 1.423 |

## 4.3  CLASSIFICATION CIFAR-10

To explore the effect of different bipolar activation functions for deep convolutional networks, we conducted some simple experiments on the CIFAR-10 dataset (Krizhevsky & Hinton, 2009) on some recent well performing architectures. We duplicated the network architectures of Oriented Response Networks (ORN) (Zhou et al., 2017) and Wide Residual Networks (WRN) (Zagoruyko & Komodakis, 2016; He et al., 2015), except we removed batch normalization, and used LSUV initialization. We then compared the performance of these networks with and without bipolar activation functions.

We used the network variants with 28 layers, a widening factor of 10, and 30% dropout, which gave the best results on CIFAR-10 in the expirements of Zagoruyko & Komodakis (2016) and Zhou et al. (2017). We also duplicated their data preprocessing, using simple mean/std normalization of the images, and horizontal flipping and random cropping as data augmentation.

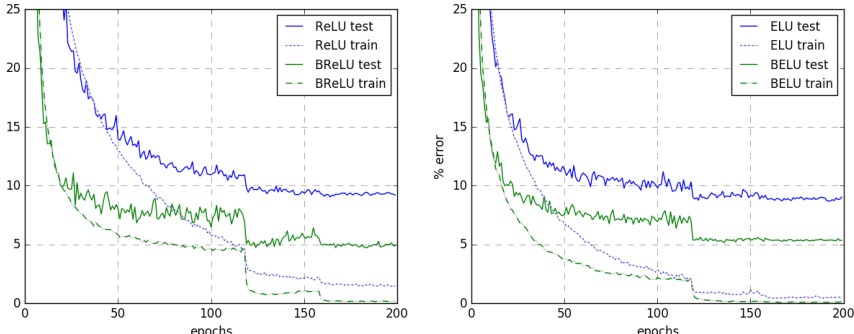

Figure 3: Training and test error on CIFAR-10, with and without bipolar units, in a 28-layer Oriented Response Network without batch normalization.

Removing batch normalization required us to lower the learning rate. For each network, we looked for the highest possible learning rate, starting at 0.08, and retrying with half the learning rate if the learning diverged. The learning rate was eased in over the first 2000 batches. The networks were trained for 200 epochs, where the learning rate was divided by five in epoch 120 and 160. Table 5 lists the test error of the last epoch for each run. The networks with bipolar activations allowed training with up to 64 times higher learning rates. As can be seen in Figure 3, the networks with bipolar activations saw a faster drop in training error, and achieved lower test errors. Note that neither setup beats the originally reported results for the networks with batch normalization (a test error of 2.98% for ORN and 4.17% for WRN).

Table 5: CIFAR-10 test error with moderate data augmentation [%]

| Network | ReLU | BReLU | ELU | BELU |
|---|---|---|---|---|
| OrientedResponseNet-28 (no BN, 30% dropout) | 9.20 | 4.91 | 9.03 | 5.35 |
| WideResNet-28 (no BN, 30% dropout) | 9.78 | 6.03 | 7.69 | 6.12 |

## 5  CONCLUSION

We have introduced bipolar activation functions, as a way to pull the mean activation of a layer towards zero in deep neural networks. Through a series of experiments, we show that bipolar ReLU and ELU units can improve trainability in deeply stacked, simple recurrent networks and in convolutional networks.

Deeply stacked RNNs with unbounded activation functions provide a challenging testbed for learning dynamics. We present empirical evidence that bipolarity helps trainability in this setting, and find that in several of the networks we trained, using bipolar versions of the activation functions was necessary for the networks to converge. These deeply stacked RNNs achieve test errors that improve upon the best previously reported results for non-gated networks on the Penn Treebank and Text8 character level language modeling tasks. Key ingredients to the model, in addition to bipolar activation functions, are residual connections, the depth of the model, LSUV initialization and proper regularization.

In our experiments on convolutional networks without batch normalization, we found that bipolar activation functions can allow for training with much higher learning rates, and that the resulting training process sees a much quicker fall in training error, and ends up with a lower test error than with their non-bipolar variants.

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

## A    SMEARED GRADIENTS IN DEEPLY STACKED RNNs

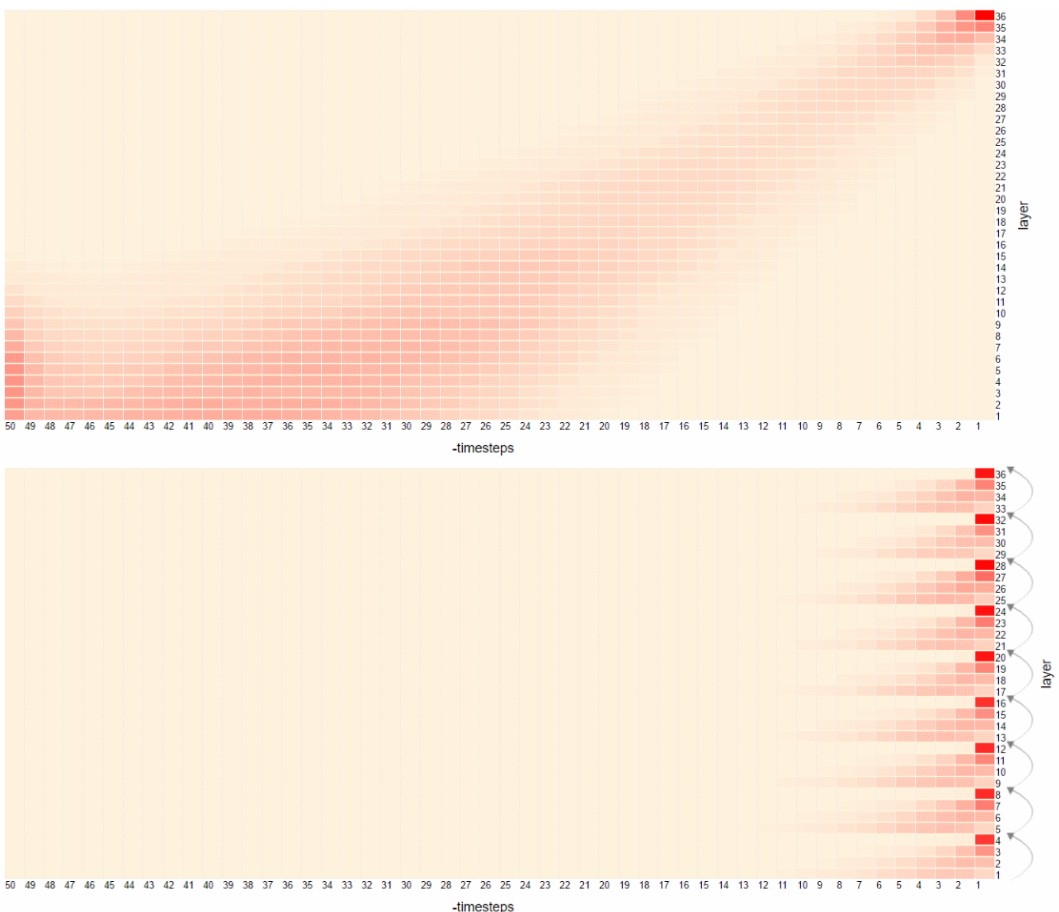

Figure 4: The L2 norm of the gradient on the output of each layer as it propagates back through time. The gradient was calculated for the last timestep per batch and backpropagated through time. This was computed for the first 10 batches in the learning process, and averaged over those. Maximal redness indicates the maximum gradient magnitude. **Top:** Without skip connections. **Bottom:** With skip connections. Both networks are 36-layer vanilla RNNs, with 256 bipolar ELU units, LSUV-initialized, trained on character-level Penn Treebank.

While LSUV initialization allows training to work in deeper stacks of RNN layers, even with LSUV we get into trouble when the stacks get deep enough (see Figure 5).

Looking at the gradient flow reveals what the problem is. When the horizontal connections $W_i$ and the vertical connections $U_i$ are of approximately equal magnitude, the gradient is distributed in equal parts vertically and horizontally. The effect of this can be seen in Figure 4 (top), where the gradient is smeared in a 45 degree angle away from its origin. This is undesirable: For example, in the 36 layer network, the error signal that reaches the first layer mostly relates to the inputs around 36 timesteps in the past.

As can be seen in Figure 4 (bottom), the problem is remedied by adding skip connections between groups of layers.

## B    TRAINING LOSS CURVES

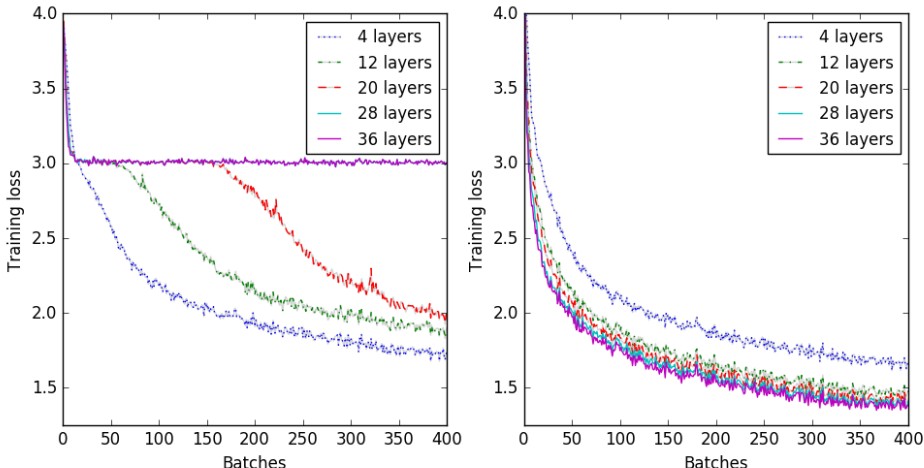

Figure 5: Skip connections help learning in deeply stacked LSUV-initialized RNNs. **Left:** Without skip connections. **Right:** With skip connections connecting groups of four layers. Both plots show training loss for vanilla RNNs with 256 bipolar ELU units per layer, LSUV-initialized, trained on character-level Penn Treebank.

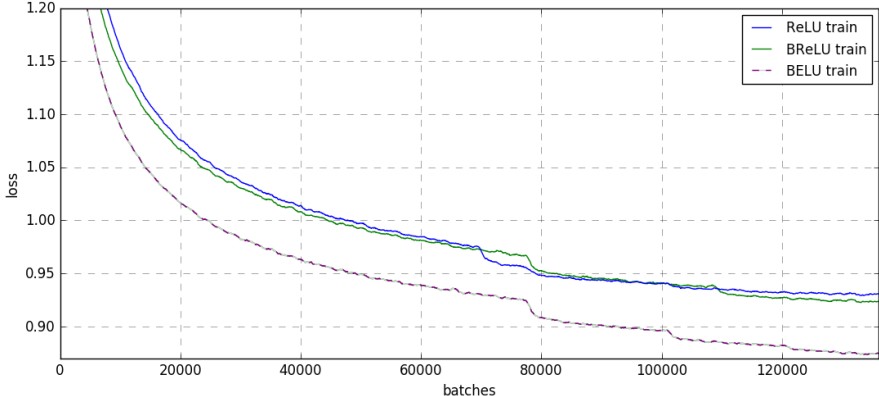

Figure 6: Training loss with various activation functions, in 36-layer RNNs on Penn Treebank. The BReLU-RNN has a lower training error than ReLU-RNN at all times where the curves are comparable (until the learning rate is cut on the ReLU-RNN). The BELU-RNN has the lowest training error of all. With the ELU-RNN, training diverged quickly.

