# OpenReview forum: "Shifting Mean Activation Towards Zero with Bipolar Activation Functions"
_ICLR.cc/2018/Conference — Invite to Workshop Track_

### Official Review · AnonReviewer2 · 2017-11-27
**Interesting paper, would like to see more experiments**

**Rating:** 4
**Confidence:** 4

**Review:**

The paper proposed a new activation function that tries to alleviate the use of  other form of normalization methods for RNNs. The activation function keeps the activation roughly zero-centered.

In general, this is an interesting direction to explore, the idea is interesting, however, I would like to see more experiments

1. The authors tested out this new activation function on RNNs. It would be interesting to see the results of the new activation function on LSTM.

2. The experimental results are fairly weak compared to the other methods that also uses many layers. For PTB and Text8, the results are comparable to recurrent batchnorm with similar number of parameters, however the recurrent batchnorm model has only 1 layer, whereas the proposed architecture has 36 layers.

3.  It would also be nice to show results on tasks that involve long term dependencies, such as speech modeling.

4. If the authors could test out the new activation function on LSTMs, it would be interesting to perform a comparison between LSTM baseline, LSTM + new activation function, LSTM + recurrent batch norm.

5. It would be nice to see the gradient flow with the new activation function compared to the ones without.

6. The theorems and proofs are rather preliminary, they may not necessarily have to be presented as theorems.

---

> ### Author Response · Authors · 2017-12-10
> **Response to AnonReviewer2**
>
> Thank you for your review.
>
> We agree that it would be nice to show results with LSTMs or GRUs. However, it is not obvious to us how to do best do so, since LSTM and GRU do not use ReLU-family activation functions, but instead use the tanh and sigmoid functions. Properly introducing bipolar activations to gated network seems to raise enough questions to warrant a paper on its own. It certainly seems like a fruitful direction for future research.
>
> Your review raises some valid concerns about our stacked RNN architecture. The many layers makes it computationally expensive, and it is outperformed by other architectures. However, our paper is fundamentally about bipolar activation functions, not about the RNN architecture.
>
> Our intent is to argue in favor of BReLU over ReLU and BELU over ELU. It is not to argue in favor of stacked Elman-RNNs over LSTMs.
>
> Most successful RNNs use gates and bounded activation functions (tanh and sigmoid). RNNs with unbounded activation functions have a potential for exploding activations. Stacking such models in depth compounds this problem, as exploding dynamics can happen depthwise as well as across time.
>
> In other words, the architecture we chose is one that makes learning hard, not one that makes it easy. The propensity for exploding dynamics makes it a good testbed for a self-centering activation function. Indeed, in more than half of our experiments on RNNs, we find that bipolar activation functions are required for the training to work at all.

---

### Official Review · AnonReviewer1 · 2017-11-27
**Nice idea but weak empirical performances**

**Rating:** 5
**Confidence:** 5

**Review:**

This paper proposes a self-normalizing bipolar extension for the ReLU activation family. For every neuron out of two, authors propose to preserve the negative inputs. Such activation function allows to shift the mean of i.i.d. variables to zeros in the case of ReLU or to a given saturation value in the case of ELU.

Combined with variance preserving initialization scheme, authors empirically observe that the bipolar ReLU allows to better preserve the mean and variance of the activations through training compared to regular ReLU for a deep stacked RNN.

Authors evaluate their bipolar activation on PTB and Text8 using a deep stacked RNN.  They show that bipolar activations allow to train deeper RNN (up to some limit) and leads to better generalization performances compared to the ReLU /ELU activation functions. They also show that they can train deep residual network architecture on CIFAR without the use of BN.

Question:
- Which layer mean and variance are reported in Figure 2? What is the difference between the left and right plots?
- In Table 1, we observe that ReLU-RNN (and BELU-RNN for very deep stacked RNN) leads to worst validation performances. It would be nice to report the training loss to see if this is an optimization or a generalization problem.
- How does bipolar activation compare to model train with BN on CIFAR10?
- Did you try bipolar activation function for gated recurrent neural networks for LSTM or GRU?
- As stated in the text, BELU-RNN outperforms BN-LSTM for PTB. However, BN-LSTM outperforms BELU-RNN on Text8. Do you know why the trend is not consistent across datasets?

-Clarity/Quality
The paper is well written and pleasant to read


- Originality:
Self-normalizing function have been explored also in scaled ELU, however the application of self-normalizing function to RNN seems novel.

- Significance:
Activation function is still a very active research topic and self-normalizing function could potentially be impactful for RNN given that the normalization approaches (batch norm, layer norm) add a significant computational cost. In this paper, bipolar activations are used to train very deep stacked RNN. However, the stacked RNN with bipolar activation are not competitive regarding to other recurrent architectures. It is not clear what are the advantage of deep stacked RNN in that context.

---

> ### Author Response · Authors · 2017-12-10
> **Response to AnonReviewer1**
>
> Thank you for your review.
>
> We address your questions and comments below:
>
> * Which layer mean and variance are reported in Figure 2? What is the difference between the left and right plots?
> - These graphs show the development of a repeated application of matrix-multiplication + non-linearity on a random vector. This is like a single layer RNN without any input and without any learning. It is an idealized case which serves to isolate the effect of the recurrent dynamics. The left graph show ReLU vs BReLU while the right graph show ELU vs BELU. In every case, the bipolar variants lead to more stable dynamics.
>
> * In Table 1, we observe that ReLU-RNN (and BELU-RNN for very deep stacked RNN) leads to worst validation performances. It would be nice to report the training loss to see if this is an optimization or a generalization problem.
> - Agreed. As noted in our reply to reviewer 3 above, we have updated the paper with a training error curve for ReLU-RNN vs BReLU-RNN, which shows lower training error with the bipolar variants.
>
> * How does bipolar activation compare to model train with BN on CIFAR10?
> - We note the original results with BN in the last sentence in the section on CIFAR-10 (a test error of 2.98% for ORN and 4.17% for WRN). These results come from an extensive hyperparameter search over networks with BN. We simply copied the hyperparameters for their best results, i.e. we have not attempted to compensate for the loss of regularization due to removing BN.
>
> * Did you try bipolar activation function for gated recurrent neural networks for LSTM or GRU?
> - It is not clear how to introduce bipolar activations to such networks, since both GRU and LSTM use only bounded activation functions like tanh and sigmoid. Most RNNs use bounded activation functions, which avoid exploding dynamics. This explosion risk makes deeply stacked RNNs with unbounded activations a good testing ground for a self-centering activation function.
>
> * As stated in the text, BELU-RNN outperforms BN-LSTM for PTB. However, BN-LSTM outperforms BELU-RNN on Text8. Do you know why the trend is not consistent across datasets?
> - We have not specifically investigated this question. A possible explanation is that we did not do any hyperparameter tuning on the Text8 dataset.
>
> * Performance of the stacked RNN
>
> It is true that other architectures outperform our stacked RNN. However, our intent was not to introduce a new RNN architecture, but to introduce bipolar activation functions. Indeed, our RNN architecture is simply a stacked Elman-RNN with skip connections.
>
> We argue that bipolar activations help learning over non-bipolar ReLU-family activations functions. To show this, we compare ReLU vs BReLU and ELU vs BELU in various architectures, and find bipolarity to be helpful in both RNNs and ConvNets. This argument does not rely on stacked RNNs beng superior to LSTMs or other recurrent architectures.

---

> > ### Comment · AnonReviewer1 · 2018-01-12
> > **Thanks for your answer**
> >
> > Thanks for your answer,
> >
> >  I like the general idea of  bipolar activation,  but I think the empirical evaluation still need to be improved. Although authors show that bipolar activation improve the trainability of deep-stack RNN and simple convolutional networks, their approach tends to underperform other methods that also focus on in networks trainability (gating, batch-norm).

---

### Official Review · AnonReviewer3 · 2017-11-28
**Simple idea to encourage zero mean activations, but results focus on accuracy instead of learning speed-up**

**Rating:** 5
**Confidence:** 3

**Review:**

Summary:
This paper proposes a simple recipe to preserve proximity to zero mean for activations in deep neural networks. The proposal is to replace the non-linearity in half of the units in each layer with its "bipolar" version -- one that is obtained by flipping the function on both axes.
The technique is tested on deep stacks of recurrent layers, and on convolutional networks with depth of 28, showing that improved results over the baseline networks are obtained.

Clarity:
The paper is easy to read. The plots in Fig. 2 and the appendix are quite helpful in improving presentation. The experimental setups are explained in detail.

Quality and significance:
The main idea from this paper is simple and intuitive. However, the experiments to support the idea do not seem to match the motivation of the paper. As stated in the beginning of the paper, the motivation behind having close to zero mean activations is that this is expected to speed up training using gradient descent. However, the presented results focus on the performance on held-out data instead of improvements in training speed. This is especially the case for the RNN experiments.

For the CIFAR-10 experiment, the training loss curves do show faster initial progress in learning. However, it is unclear that overall training time can be reduced with the help of this technique. To evaluate this speed up effect, the dependence on the choice of learning rate and other hyperparameters should also be considered.

Nevertheless, it is interesting to note the result that the proposed approach converts a deep network that does not train into one which does in many cases. The method appears to improve the training for moderately deep convolutional networks without batch normalization (although this is tested on a single dataset), but is not practically useful yet since the regularization benefits of Batch Normalization are also taken away.

---

> ### Author Response · Authors · 2017-12-10
> **Response to AnonReviewer3**
>
> Thanks for your review. It is a fair criticism that we have not included enough evidence of faster training in the RNN setting.
>
> What follows is a summary of the evidence we do present that bipolar activations help learning in RNNs. There are two cases to consider: ELU vs BELU and ReLU vs BReLU.
>
> - For the ELU vs BELU case, we find that in every experiment the ELU-RNN diverges, while the BELU-RNN does not.
> - For the ReLU vs BReLU case, we find that the ReLU-RNN diverges in the Text8 experiment, while BReLU-RNN does not.
>
> In most of our experiments, the non-bipolar RNNs do not converge at all, while the bipolar variant does.
>
> However on PennTreebank, both ReLU-RNN and BReLU-RNN do converge. Here the bipolar version achieves higher generalization accuracy on deeper models. This higher accuracy may be because bipolarity helps optimization, and it may be because of better generalization when using the bipolar versions. It is right to point out that the paper does not adequately establish which of these two are happening.
>
> The way to establish that the PennTreebank results are due to ease of optimization would be to present the training error curve for the two variants. While we don't present it in the paper, we do have this curve. For the 36-layer network we focus on, what it shows is that the bipolar variant has lower training error for the first 88 epochs, until the learning rate is cut in the ReLU-RNN. In other words, at every point where the curves are comparable, the BReLU-RNN error is lower than the ReLU-RNN error, and the BReLU-RNN also ends up with a lower training error in the end.
>
> That these curves are not in the paper is an omission, and we have updated the paper to include it. Even without this curve, we believe that the remaining evidence makes a strong case that bipolar activations help learning:
>
> - With ConvNets on CIFAR-10, the bipolar version achieved substantially lower training error than the non-bipolar versions.
> - On Text8, both non-bipolar version diverge, while the bipolar versions do not.
> - On PennTreebank, the ELU diverges, while the BELU does not.
>
> For the remaining case, BReLU vs ReLU on PennTreebank we have updated the paper with the learning curve that shows faster learning in the bipolar case.

---

### Public Comment · ~Andrew_Brock1 · 2018-03-13
**Similarity to NCReLU**

Can the authors comment on the similarity to NCReLU from V1 of the DiracNets paper (https://arxiv.org/abs/1706.00388v1, June 2017)? Their functional form appears nearly identical but for the fact that NCReLU duplicates the signal (which is important but still merits comparison).

---

> ### Author Response · Authors · 2018-03-14
> **Re: NCReLU**
>
> Indeed, the form proposed in this paper is very similar, thanks for bringing it to our attention (version 1, it appears to be removed in v2). The difference is, as you say, that the NCReLU duplicates the neuron population. Because the weights are not duplicated (but recieve gradients from each copy), the learning dynamics would be different in such a network. For what it is worth, we submitted our paper to NIPS in May of 2017, this appears to be a concurrent development.

---

### Decision · Program_Chairs · 2018-01-29
**ICLR 2018 Conference Acceptance Decision**

**Decision:**

Invite to Workshop Track

**Comment:**

the reviewers were not fully convinced of the setting under which the proposed bipolar activation function was found by the authors to be preferable, and neither am i.